# Immunomonitoring via ELISPOT Assay Reveals Attenuated T-Cell Immunity to CMV in Immunocompromised Liver-Transplant Patients

**DOI:** 10.3390/cells13090741

**Published:** 2024-04-24

**Authors:** Ann-Kristin Traska, Tobias Max Nowacki, Richard Vollenberg, Florian Rennebaum, Jörn Arne Meier, Tina Schomacher, Sara Noemi Reinartz Groba, Julia Fischer, Jonel Trebicka, Phil-Robin Tepasse

**Affiliations:** 1Department of Medicine B for Gastroenterology, Hepatology, Endocrinology and Clinical Infectiology, University Hospital Münster, 48149 Münster, Germany; ann-kristin.traska@ukmuenster.de (A.-K.T.); richard.vollenberg@ukmuenster.de (R.V.); florian.rennebaum@ukmuenster.de (F.R.); joernarne.meier@ukmuenster.de (J.A.M.); tina.schomacher@ukmuenster.de (T.S.); saranoemi.reinartzgroba@ukmuenster.de (S.N.R.G.); julia.fischer2@ukmuenster.de (J.F.); jonel.trebicka@ukmuenster.de (J.T.); 2Department of Internal Medicine and Gastroenterology, Marienhospital Steinfurt, 48565 Steinfurt, Germany; tobias.nowacki@ukm-mhs.de

**Keywords:** liver transplantation, cytomegalovirus, CMV, opportunistic infection, immunosuppressive therapy

## Abstract

Assessing immune responses to cytomegalovirus (CMV) after liver transplant in patients on immunosuppressive therapy remains challenging. In this study, employing ELISPOT assays, 52 liver-transplant recipients were evaluated for antiviral T-cell activity in peripheral blood mononuclear cells (PBMCs), measuring interferon-γ (IFN-γ) secretion upon stimulation with CMV-specific peptides (CMV peptide pool, CMV IE-1, and pp65 antigens). Parameters such as stimulation index, mean spot size, and mean spot count were measured. The study found that heightened immunosuppression, especially with prednisolone in triple therapy, significantly dampened CMV-specific immune responses. This was demonstrated by decreased IFN-γ production by CMV-specific T-cells (CMV peptide pool: *p* = 0.036; OR = 0.065 [95% CI: 0.005–0.840], pp65 antigen: *p* = 0.026; OR = 0.048 [95% CI: 0.003–0.699]). Increased immunosuppression correlated with reduced IFN-γ secretion per cell, reflected in smaller mean spot sizes for the CMV peptide pool (*p* = 0.019). Notably, shorter post-transplant intervals correlated with diminished antiviral T-cell IFN-γ release at two years (CMV peptide pool: *p* = 0.019; IE antigen: *p* = 0.010) and five years (CMV peptide pool: *p* = 0.0001; IE antigen: *p* = 0.002; pp65 antigen: *p* = 0.047), as did advancing age (pp65 antigen: *p* = 0.016, OR = 0.932, 95% CI: 0.881–0.987). Patients with undetectable CMV antigens had a notably higher risk of CMV reactivation within six months from blood collection, closely linked with triple immunosuppression and prednisolone use. These findings highlight the intricate interplay between immunosuppression, immune response dynamics, and CMV reactivation risk, emphasizing the necessity for tailored immunosuppressive strategies to mitigate CMV reactivation in liver-transplant recipients. It can be concluded that, particularly in the early months post-transplantation, the use of prednisolone as a third immunosuppressant should be critically reconsidered. Additionally, the use of prophylactic antiviral therapy effective against CMV in this context holds significant importance.

## 1. Introduction

Human cytomegalovirus (CMV) belongs to the human herpesvirus family and is highly prevalent worldwide. In adults, the seroprevalence of CMV is often above 50% in many European countries, and in some age groups, it can even reach or exceed 90%. It is crucial to note that the precise figures may vary depending on the study, region, and population group [1,2]. CMV enters a latency phase following the initial infection, often persisting asymptomatically in tissues or peripheral blood mononuclear cells throughout an individual’s life [3,4]. In rare instances, CMV infections may lead to severe infections in immunocompetent individuals, while this persistence remains uncomplicated for most immunocompetent individuals. Patients undergoing immunosuppressive therapy are at a relevant risk of CMV reactivation [5]. CMV reactivation represents one of the most common opportunistic infections after liver transplantation and is frequently associated with severe morbidity and mortality [6]. Moreover, CMV infection has been linked to an increased risk of acute or chronic graft rejection [7]. Additionally, CMV reactivation has been associated with transplanted organ thrombosis, reactivation of the hepatitis C virus, and an elevated risk of other opportunistic infections owing to immune system damage [8,9,10]. Some subgroups of liver-transplant patients appear to have a higher risk of CMV reactivations. This has been demonstrated, for example, in PSC (Primary Sclerosing Cholangitis) patients [11]. A reliable diagnosis, particularly with predictive value regarding the occurrence of a CMV infection, is of great importance. Available detection methods, such as interferon-gamma ELISPOT (Enzyme-Linked Immunosorbent Spot) assays and interferon-gamma ELISA (Enzyme-Linked Immunosorbent Assay), exhibit varying diagnostic values [12,13].

In this study, we employed an ex vivo single-cell resolution ELISPOT assay to measure antigen-specific secretion of interferon-γ (IFN-γ) to evaluate antiviral T-cell immunity in a cohort of 52 liver-transplant recipients who were serologically positive for CMV. The aim of our study was to investigate the effect of different immunosuppressive therapeutic strategies on antiviral cellular immune responses toward CMV in patients after liver transplantation. Since CD8^+^ -T-cells play a crucial role in (long-term) viral control, we focused on measuring T-cell-derived immune response by measuring antigen-specific secretion of IFN-γ in ELISPOT assays, which have been shown to allow for the identification of virus-specific T-cells that had been involved in immune surveillance in vivo [14,15,16]. 

## 2. Materials and Methods

### 2.1. Patient Cohort and Peripheral Blood Mononuclear Cell (PBMC) Sampling

The patient cohort used in this study comprised 52 adult individuals aged from 26 to 80 years. All participants gave informed consent to be included in the study, had undergone at least one liver transplantation, and were confirmed to be seropositive for human cytomegalovirus (CMV) at the time of sample acquisition. Detailed information concerning the patient collective can be found in Table 1. Besides analyzing patient groups affected by different immunosuppressive agents, we also compared these patients to a healthy control group (*n* = 7) not on any immunomodulating medications. Serological testing for CMV was performed as part of the liver transplantation listing evaluation, following standard protocols established by the Department of Clinical Virology at the University Hospital Muenster. Peripheral Blood Mononuclear Cells (PBMCs) were collected by obtaining EDTA blood from each subject, and PBMCs were isolated using standard laboratory protocols as previously reported [17]. Cryopreservation was then conducted, ensuring the preservation of lymphocyte functionality upon thawing in accordance with established protocols [18]. For the detection of CMV reactivation post-transplantation, the liver-transplant outpatient clinic conducts regular CMV PCR testing from EDTA blood as a standard of care examination for all patients every three months. This study received approval from the Local Ethics Committee of the University Hospital Münster, adhering to ethical guidelines and ensuring patient confidentiality and welfare throughout the research process (file: 2020-566-f-S).

### 2.2. Thawing and Preparation of Cells

Upon testing, PBMCs were thawed as previously described [19]. Briefly, cryovials containing the PBMCs were subjected to a controlled increase in temperature in a water bath until completely thawed. Subsequently, the cell suspension was diluted using cRPMI (Bio-Whittaker, Walkersville, MA, USA) solution. The cell count and vitality were then determined under a UV microscope with Acridine Orange/Ethidium Bromide (Sigma–Aldrich, St. Louis, MO, USA/Fisher Scientific, Pittsburgh, PA, USA) staining. Before plating the PBMCs for further experimentation, the samples’ PBMC concentration was adjusted to 4 million cells per milliliter by adding cRPMI supplemented with fresh Glutamine (Gibco BRL, Grand Island, NY, USA), according to the previously counted cell concentration. This ensured uniformity in cell density and facilitated accurate and standardized analysis in subsequent assays.

### 2.3. Antigens

For this study, T-cell activation was performed in vitro using a CMV-MHC Class I Control Peptide Pool containing five HLA class I restricted T-cell epitopes from CMV (Cellular Technology Limited, Shaker Heights, OH, USA). The following peptides are contained in the pool: NLVPMVATV (CMV, HLA-A2), SDEEEAIVAYTL (CMV, HLA-B18), IPSINVHHY (CMV, HLA-B35), EFFWDANDIY (CMV, HLA-B44), and TPRVTGGGAM (CMV, HLA-B7). The CMV-MHC Class I Control Peptide Pool comprises five peptides, each corresponding to a defined HLA class I restricted T-cell epitope from cytomegalovirus. These peptides, initially described in CMV AD169 (ATCC VR-538) and CMV RVAd65 strains by Wills et al., have been demonstrated to elicit recall responses in individuals expressing these commonly found MHC class I alleles [20]. Previous studies have indicated that the majority of randomly selected human donors respond to the CMV-MHC Class I Control Peptide Pool [21].

Additionally, T-activated^®^ immunodominant CMV IE-1 and pp65 antigens (Lophius Biosciences GmbH, Regensburg, Germany) which encompass the immunodominant region of the CMV pp65 (amino acids 366 to 546, CMV strain AD169) and the full-length IE-1 (amino acids 1–491, CMV Towne strain) were utilized as already described [22,23,24].

### 2.4. Human Interferon-y Immunospot Assay

In this study, the Human Interferon-γ Single Color ImmunoSpot Assay test kit from CTL (Cellular Technology Limited, Shaker Heights, OH, USA) was utilized according to the manufacturer’s instructions [25]. Briefly, the PVDF membrane of the plate was precoated with a combination of Human IFN-γ Capture Solution, consisting of a 1:250 ratio of Human Interferon-γ Capture Antibody diluted in the solvent provided by the manufacturer (Cellular Technology Limited, Shaker Heights, OH, USA) and sterile tissue culture tested PBS (phosphate buffered saline; Lonza, Walkersville, MD, USA), which served as a buffer. The plates were incubated overnight in a humidified box at 4 °C. Subsequently, the plate was washed and blocked using PBS-BSA (phosphate buffered saline–bovine serum albumin; Lonza, Walkersville, MD, USA/Sigma–Aldrich Chemie GmbH, Taufkirchen, Germany). CMV antigens were added to the wells in constellations previously established as optimal to induce immune responses, namely 1:10 dilution of cRMPI and the CMV pool stock solution provided by the manufacturer for the CMV pool and 1:25 dilution of cRMPI and the provided antigen stock solutions for IE antigen and pp65, respectively. Thawed PBMCs were then added to the wells at a cell suspension of 100 µL per well containing 2 × 10^5^ cells/well. The plates were incubated in a humidified CO_2_ incubator at 37 °C for 24 h. On the following day, the plates were washed using sterile tissue culture tested PBS and PBS-TWEEN-BSA (phosphate buffered saline-polysorbate-bovine serum albumin; Lonza, Walkersville, MD, USA/Sigma–Aldrich Chemie GmbH, Taufkirchen, Germany), after which the Anti-human IFN-γ (Biotin) Detection Antibody solution, consisting of a 1:250 ratio of Anti-human IFN-γ Detection Antibody diluted in the solvent provided by the manufacturer (Cellular Technology Limited, Shaker Heights, OH, USA) was added to the wells. The plates were then stored at room temperature for two hours. To complete the spot formation process, Streptavidin alkaline phosphatase dilution (Cellular Technology Limited, Shaker Heights, OH, USA) was added to each well after washing the plates. Following a 30-minute incubation period, the plates were washed with PBS-TWEEN-BSA and emptied. To initiate color development, 80 µL of Blue Developer Solution was added per well, consisting of 10 mL of Diluent Blue combined with three substrate solutions, all provided by the manufacturer (Cellular Technology Limited, Shaker Heights, OH, USA). The reaction was stopped after 15 min by rinsing the plates with tap water, and the plates were left to airdry overnight. ELISPOT plates were scanned and analyzed using an ImmunoSpot^®^ Reader by CTL. Spots were counted automatically using the ImmunoSpot^®^ 5.041 Software (Basic Count™ mode) for each antigen stimulation condition and the medium negative controls. The counted spots were reported as mean spot-forming units (SFU) per well of duplicate wells (±SD). Additionally, spot size was determined using the inbuilt ImmunoSpot Software as a measure of the amount of IFN-γ secreted. 

### 2.5. Statistical Analysis of Immunospot Assay

The stimulation index (SI) was used to quantify the frequency of T-cells showing antigen-specific IFN-γ secretion as previously described [26]. The SI was calculated by dividing the spot-forming units (SFU) of the specific antigens by the SFU of the corresponding negative control. Immune responses towards an antigen were considered positive if the number of spots in the antigen wells was at least thrice the number of spots generated with medium alone (i.e., SI ≥ 3).

In the case of non-parametric data distribution in the analysis of clinical parameters, the Kruskal–Wallis Test and pairwise comparisons using Bonferroni correction were used for the comparison of multiple groups and the analysis of two independent groups, respectively. Additionally, the Mann–Whitney U Test was employed when comparing the two groups. In the case of parametric data distribution in the analysis of clinical parameters, ANOVA and *t*-test (Welch) were used. The Pearson Chi-Square Test or the exact Fisher Test was utilized as appropriate for analyzing categorical variables. A *p*-value of <0.05 was considered statistically significant. The computer programs SPSS version 23 (IBM, Armonk, NY, USA) and SigmaPlot version 11 (Systat Software Inc., San Jose, CA, USA) were used for statistical analyses.

## 3. Results

### 3.1. Immunosuppressive Therapy Reduces Frequency of CMV Reactive T-Cells

General patient characteristics are shown in Table 1. Patients received an individually tailored immunosuppressive therapy based on their comorbidity status and various intolerances, as determined by their physician. In addition to common immunosuppressants such as tacrolimus, mycophenolate mofetil, everolimus, and cyclosporine, eight patients were prescribed prednisolone. Among them, seven patients received prednisolone as the third immunosuppressive agent within the first months post-transplantation as part of standard care treatment. Patients receiving single, dual, or triple immunosuppressive therapies were compared to analyze the strength of immunosuppression on the frequency of CMV reactive T-cells. T-cell responses to two of the three tested antigens were significantly reduced in patients receiving more than one immunosuppressive therapy. The frequency of CMV reactive T-cells was significantly reduced in patients on triple immunosuppression compared to both dual (*p* = 0.017) and single therapy (*p* = 0.005) when evaluating the CMV peptide pool. Patients with triple immunosuppressive treatment exhibited a mean stimulation index (SI) of 6.32 (Inter Quartile Range (IQR) 0–4.71) in response to the CMV peptide pool. In contrast, those on dual immunosuppression demonstrated a mean SI of 124.19 (IQR: 1.91–191.50), while patients with single immunosuppression displayed a mean SI of 179.6 (IQR: 2.82–308.75). Similarly, a significant decrease in SI values on triple immunosuppression was detected in response to the IE antigen as compared to single (*p* = 0.007) immunosuppression. Patients with triple immunosuppression yielded a mean SI of 2.3 (IQR: 0.06–5.0), whereas dual therapy and single therapy resulted in SIs of 53.05 (IQR: 1.18–28.50) and 48.68 (IQR: 4.07–65.98), respectively. In response to pp65, SI values ranged from 81.25 (IQR: 1.62–97.74) for single immunosuppressant therapy to 71.97 (IQR: 0.38–78.25) for dual therapy and 2.94 (IQR: 0–2.11) for triple immunosuppressive therapy. However, this trend was not significant (Figure 1A–C). Healthy control subjects demonstrated no differences in the mean SI compared to patients on one immunosuppressive agent when evaluating the CMV peptide pool (*p* = 0.22), the IE antigen (*p* = 0.67), and the pp65 antigen (*p* = 0.10).

### 3.2. Immunosuppressive Therapy Reduces IFN-γ Release of Antiviral T-Cells

Significant differences were detected when evaluating the amount of IFN-γ produced per single cell, measured as spot size, and the number of immunosuppressants applied.

For the CMV pool, a mean spot size of 6.75 10^−3^ mm^2^ (IQR: 0.00–10.86 10^−3^ mm^2^) was detected in patients on triple immunotherapy. Patients on dual therapy displayed a mean spot size of 10.83 10^−3^ mm^2^ (IQR: 8.80–13.85 10^−3^ mm^2^), while a mean spot size of 12.68 10^−3^ mm^2^ (IQR: 11.02–14.46 10^−3^ mm^2^) was measured in patients receiving monotherapy. Comparing these values, the spot size decreased significantly with an increasing number of immunosuppressants applied (*p* = 0.019). Comparing the MSS of healthy controls to patients on one immunosuppressive agent, no differences were found (*p* = 0.24) (Figure 1D and Figure 2).

### 3.3. Immunosuppressive Therapy Reduces the Total Number of Detected Experimental Antigen Approaches

A total of three experimental antigen approaches were investigated (Section 2). When measuring the number of antigen approaches (pp65, IE, or peptide pool) that elicited a strong immune response, it was observed that healthy controls and patients on monotherapy exhibited a higher antigen detection rate than those on triple therapy. Within the triple immunosuppressant group, all three antigen approaches were detected in 11.1% of the patients, while 77.8% of the patients showed no antigen detection. Among patients receiving two immunosuppressants, 40.7% displayed the detection of three antigen approaches, whereas 68.8% of those on monotherapy detected all three antigen approaches. Statistical analysis using the Pearson Chi-Square Test revealed significantly higher antigen detection rates for patients on monotherapy across all tested antigens (CMV peptide pool: *p* = 0.025; IE antigen: *p* = 0.037; pp65: *p* = 0.019). For healthy controls in comparison to patients on one immunosuppressive agent, no differences in the total number of detected experimental antigen approaches were found in the exact Fisher Test performed (CMV peptide pool: *p* = 0.63; IE antigen: *p* = 1.00; pp65: *p* = 0.62) (Table 2).

### 3.4. Time Interval after Transplantation Correlates with Reduced IFN-γ Release of Antiviral T-Cells 

Significant differences were observed when comparing the mean spot size in patients regarding the time that had elapsed since transplantation. Generally, patients had larger spot sizes, which took longer since transplantation. In patients whose transplantation was at least two years ago (*n* = 41), the mean spot size in response to the CMV peptide pool antigen was 11.56 10^−3^ mm^2^ (IQR: 9.47–14.41^−3^ mm^2^) vs. 7.49 10^−3^ mm^2^ (IQR: 3.23–9.38 10^−3^ mm^2^) in those whose transplantation was less than two years ago (*n* = 11) (*p* = 0.019). A similar result was observed when comparing the data for patients whose transplantation was at least five years ago (*n* = 23) (Mean Spot Size (MSS): 13.41 10^−3^ mm^2^ (IQR: 11.01–15.81 10^−3^ mm^2^)) and patients with less than five years after transplantation (*n* = 29) (MSS: 8.54 10^−3^ mm^2^ (IQR: 4.00–11.86 10^−3^ mm^2^)) (*p* = 0.0001).

Significant differences were also detected in spot sizes in response to the IE antigen. Mean Spot Size was larger in patients with more than two years since transplantation (13.55 10^−3^ mm^2^ (IQR: 10:39–17.40 10^−3^ mm^2^)) as compared to patients who received a transplant within two years (8.17 10^−3^ mm^2^ (IQR: 0.00–12.17 10^−3^ mm^2^)) (*p* = 0.010). Five years post-transplantation, patients had significantly larger spot sizes as compared to patients within five years post-transplantation (15.34 10^−3^ mm^2^ (IQR: 12.58–18.04 10^−3^ mm^2^) vs. (10.08 10^−3^ mm^2^ (IQR: 5.05–15.07 10^−3^ mm^2^) (*p* = 0.002).

For pp65, comparable results were observed for at least five years post-transplantation but not for two years. Patients whose transplantation was at least five years ago showed significantly larger spots (*p* = 0.047) with a mean spot size of 13.31 10^−3^ mm^2^ (IQR: 7.96–17.90 10^−3^ mm^2^) than those who did receive their transplant within the last five years prior to sample acquisition (9.26 10^−3^ mm^2^ (IQR: 1.75–14.57 10^−3^ mm^2^)).

Examining the specified post-transplantation time intervals, significant differences emerge in the number of administered immunosuppressants across corresponding subgroups. Patients transplanted within the last two years notably receive a higher number of immunosuppressants than those transplanted more than two years ago (*p* = 0.013). Similarly, a comparable pattern is observed when comparing patients transplanted more than five years ago with those who have not undergone transplantation (*p* = 0.001).

Figure 3 displays the mean spot size (MSS) of individual antigens in relation to the time elapsed post-transplantation. Referring to all antigens, the MSS consistently increases with the passing of time after transplantation.

### 3.5. Application of Prednisolone and Increasing Age Reduce CMV-Specific T-Cell Response

A multivariate analysis was conducted to comprehensively examine the impact of various patient-specific parameters on the observed immune response. This analysis encompassed age, time since transplantation in months, presence of renal disease, and the use of prednisolone or tacrolimus in relation to antigen detection. The logistic regression analysis focused on antigen detection within the CMV pool, yielding a model summary Nagelkerkes R² of 0.377. Notably, the coefficient table (Table 3) indicated a significant association for prednisolone administration (*p* = 0.036). Prednisolone administration exhibited a negative correlation with antigen recognition, as reflected by an Odds Ratio of 0.065 (95% CI: 0.005–0.840). 

In terms of the pp65 antigen, the regression analysis (Table 3) identified significant associations with age (*p* = 0.016) and prednisolone administration (*p* = 0.026), yielding a Nagelkerkes R² of 0.309. The Odds Ratio for age (0.932, 95% CI: 0.881–0.987) indicated a negative effect of increasing age on pp65 antigen detection, with a 6.8% decreased likelihood of detection per year. Similarly, prednisolone administration displayed a negative correlation with antigen detection, reflected by an Odds Ratio of 0.048 (95% CI: 0.003–0.699). However, no significant correlations were observed for IE antigen detection (Table 3).

### 3.6. Appearance of CMV Reactivation during the Subsequent Clinical Course

In addition, an analysis of patient records was conducted regarding the reactivation of a CMV infection at any time point based on the detection of CMV DNA in patient blood (analysis as part of the standard of care provided to patients in the transplant outpatient clinic). In 10 out of 52 cases (19.2%), CMV reactivation was documented in the medical records. The reactivation was documented at a median interval of 6 months (IQR 1–26 months) from the study blood sample collection. The D+/R− CMV serostatus differed significantly between patients who experienced CMV reactivation and patients who did not (40% vs. 4.8%, *p* = 0.002). Out of the 42 patients who did not experience CMV reactivation, two were still receiving prophylactic treatment with valganciclovir at the time of the study. Of the 10 patients who experienced CMV reactivation, three were still undergoing prophylactic treatment with valganciclovir at the time of reactivation. CMV reactivation was asymptomatic in 7 out of 10 cases, while in 3 out of 10 cases, it resulted in enteritis. In all three cases, the patients developed diarrhea, and endoscopic confirmation of colitis, as well as CMV detection through PCR from colon biopsies, was performed. 

These results were subsequently compared among patients whose ELISPOT examinations detected one, two, or all three antigens. In the group where the ELISPOT analysis did not detect any antigens, 8 out of 16 patients (50%) experienced a CMV infection reactivation. Among those in whom one antigen was detected in the ELISPOT analysis, 2 out of 7 patients (28.6%) experienced reactivation. For those in whom 2 or 3 antigens were detected, no CMV reactivation occurred. In the following, subsequent analyses were done to compare the 10 individuals with CMV reactivation to 42 without, with the additional data presented in Table 4. Our supplementary findings revealed significant associations between CMV reactivation and the use of three immunosuppressive agents, as well as the administration of prednisolone. Furthermore, immune response parameters—specifically stimulation index, mean spot count, and mean spot size—demonstrated significant differences between the two groups across various CMV peptides (CMV peptide pool, IE Ag, and pp65). The detailed results of these additional analyses are summarized in Table 4.

## 4. Discussion

Our study demonstrates a significant decline in T-cell activity, assessed by IFN-γ secretion through the ELISPOT assay, with a corresponding correlation to the number of administered immunosuppressants. This reduction in cellular immune response towards CMV manifests across three T-cell immunity parameters. First, patients undergoing dual or triple immunotherapy exhibit markedly lower frequencies of antigen-specific T-cells compared to those on monotherapy [27]. Second, single-cell resolution ELISPOT measurements demonstrate smaller spots, indicating reduced IFN-γ release in individuals receiving multiple immunosuppressive drugs [28]. Thirdly, the number of detected antigens decreases significantly with the intensity of immunosuppression.

Importantly, a negative correlation is identified between antigen recognition and the additional use of prednisolone, in almost every case employed as the third immunosuppressive agent in the immunosuppressive regimens in our study [29]. Despite its role in preventing rejection, especially in the first year after transplantation [30,31], prednisolone is associated with significant adverse effects, including the development of osteoporosis, diabetes, and elevated risk of severe infections like CMV reactivation [31,32,33,34,35]. Patient record analysis reveals substantial differences in immune response parameters between patient groups with and without CMV reactivation. Notably, specific metrics such as stimulation index, mean spot count, and mean spot size showed significant distinctions. Furthermore, strong associations have been uncovered between CMV reactivation and the use of three immunosuppressive agents, along with the consistent administration of prednisolone, a third immunosuppressive agent [29,31]. It is noteworthy that demographic and clinical variables such as age, gender, and months after transplantation do not exhibit statistically significant differences. These detailed analyses underscore the crucial role of specific immunosuppressive regimens and immune response metrics in understanding CMV reactivation post-transplantation.

CMV reactivations can lead to multiple organ involvements, such as severe retinitis, enteritis, colitis, pneumonia, and encephalitis. Ultimately, the development of hepatitis with graft loss is also possible [36]. Considering the results of our study, it is crucial, especially for patients undergoing combination immunosuppressive therapy with steroids within the initial year post-transplantation, to adopt a vigilant approach by closely monitoring CMV infection. In this context, incorporating regular assessments of T-cell immunity (such as antigen-specific ELISPOT assays with established outcome measures, e.g., stimulation indices) could aid in identifying patients at risk of developing opportunistic infections. 

Our study reveals a negative correlation between the age of patients and antigen detection. There is evidence that advancing age is linked to immunosenescence, which leads to a decline in immune response [37,38]. Specifically, the quantity of CMV reactive T-cells decreased with age, aligning with the reduced immune response observed in our results [39,40]. Therefore, more intensive monitoring of T-cell activity in older patients in the years following transplantation should be considered. This heightened vigilance can aid in the timely detection and management of potential immune-related complications in aging transplant recipients.

Discrepancies were observed in the results between different antigens, particularly the pp65 antigen, compared to the CMV pool and IE antigen. This discrepancy may be attributed to various factors related to the immune response of T-cells recorded as SI. While IE1 and pp65 antigens target various CMV reactive cells, including T helper cells, cytotoxic T-cells, and natural killer cells, the peptide pool primarily targets CD8+ cytotoxic cells. However, this alone does not fully explain the observed discrepancy between pp65 and the CMV pool and IE antigen results. Pp65 is an immunodominant antigen of cytomegalovirus, but individuals may not react to this specific antigen even if infected [41]. Terlutter et al. reported the absence of an immune response for pp65 in their ImmunoSpot assay while observing immune responses for other tested antigens [17]. Moreover, the immunogenicity of IE and pp65 antigens may vary depending on the time after liver transplantation and the type of immunosuppression used. Additionally, antigen recognition in cryopreserved PBMCs or T-cells isolated from cryopreserved PBMCs can be a concern. Nevertheless, literature data and our own experience indicate maintained functionality of cryopreserved cells in ELISPOT assays when isolated according to established protocols [18]. Notably, antigen recognition might vary because of differences in peptide sequence between different CMV strains. Previous studies have shown that CMV-positive test results using the CMV pp65 and IE antigens of this study were observed in 97% CMV-seropositive healthy donors and 90% hemodialysis patients [24,42].

In our study, samples were obtained at different time intervals after transplantation. The samples from patients receiving 3 and 2 immunosuppressants were significantly collected earlier post-transplantation than those from patients on a single immunosuppressive agent. This discrepancy is due to the required intensified immunosuppressive regimen in the initial post-transplantation months, leading to the absence of samples from patients with only one immunosuppressive agent shortly after transplantation. Differences in the timing of sample collection post-transplantation may contribute to variations in the measured immune response and can influence the results regarding the impact of the number of administered immunosuppressants. Complementing these analyses, this study expanded the comparison of the T-cell immunity parameters, such as antigen detection methods, mean spot size (MSS), and stimulation index (SI), between patients undergoing immunosuppressive monotherapy and a healthy control group. Further examination of INF-γ secretion per cell, quantified by MSS, across the three antigen approaches showed no significant differences. Similarly, in assessing absolute antigen recognition, no significant discrepancies were found between the monotherapy patients and the healthy controls across all tested antigens. Additionally, the evaluation of the SI for these antigen approaches also revealed no significant variances between the two groups. In summary, these data suggest that patients receiving only a single immunosuppressant after liver transplantation undergo immune reconstitution, therefore exhibiting a similar immunity to CMV as seen in healthy control subjects. It must be noted as a limitation that our study design cannot definitively differentiate between the influence of time post-transplantation and the number of immunosuppressants. A prospective approach would be necessary for this, where patients are adjusted to 1, 2, or 3 immunosuppressants at comparable time points after transplantation.

In addition to antigen-related factors, the structure of our patient population is an important consideration. The limited size of the collective, consisting of 52 patients, and the unequal distribution of subgroups receiving one to three immunosuppressants could have influenced the study outcomes. Furthermore, the correlation of ELISPOT data with the occurrence of CMV reactivation was assessed through retrospective analyses, making it susceptible to bias. To address these limitations and draw more comprehensive conclusions, future studies should aim for larger and more balanced patient populations with longer observation periods to gain a more detailed understanding of the immune response and antigen detection in the context of immunosuppression after transplantation. Additionally, our data have implications for immunosuppressed patients in general, e.g., in the setting of other solid organ transplantations, bone marrow transplantation, HIV, cancer, and autoimmune diseases, in all of which patient care critically depends on sensitive and reliable measurements of T-cell immunity towards opportunistic viral infections. 

## 5. Conclusions

In conclusion, our study reveals a significant attenuation of CMV-specific T-cell responses in liver-transplant recipients with increasing levels of immunosuppression, as demonstrated by reduced IFN-γ production and antigen detection. Notably, patients on multiple immunosuppressive agents exhibit a pronounced decline in antiviral T-cell activity. Age and steroid use within the first post-transplantation year emerge as crucial factors influencing the reduction in T-cell immunity against CMV. Importantly, the absence of CMV antigen detection in ELISPOT analysis correlates with a higher incidence of CMV reactivation within a median of 6 months, underscoring the need for vigilant monitoring. Our findings emphasize the importance of ELISPOT analysis in assessing CMV-specific T-cell responses and advocate for regular re-evaluation of immunosuppression to maintain effective antiviral immunity.

## Figures and Tables

**Figure 1 cells-13-00741-f001:**
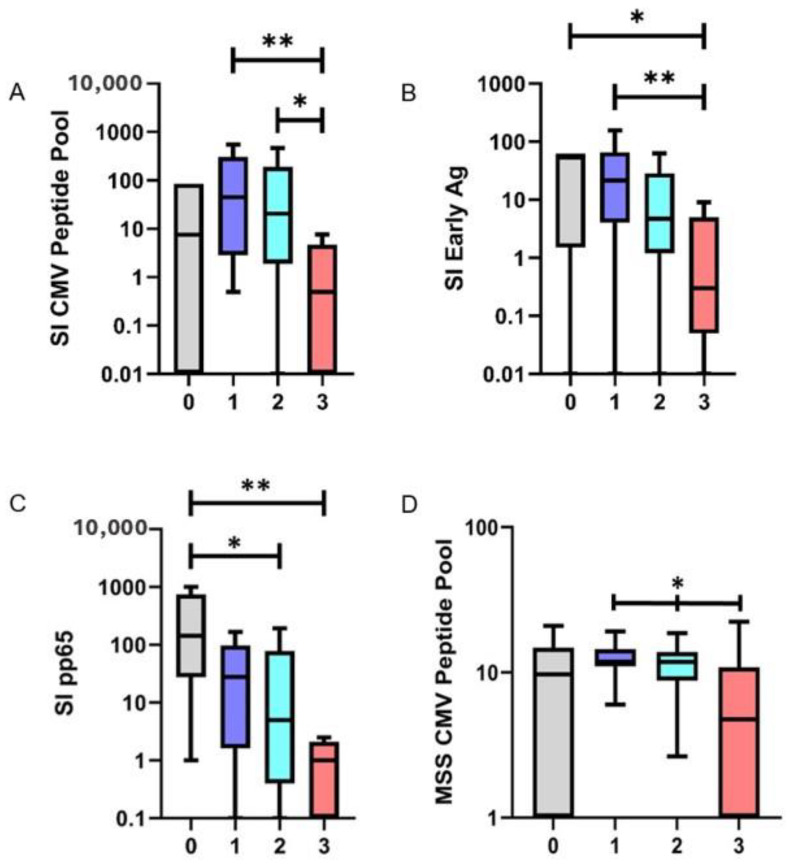
Stimulation Index (SI) of the tested antigens based on the number of administered immunosuppressants (*x*-axis) (**A**–**C**); IFN-γ release from antiviral T-cells, represented as mean spot size (MSS), in relation to the CMV peptide pool, corresponding to an escalation in the number of administered immunosuppressants (**D**). CMV = Cytomegalovirus, Ag = Antigen; ** *p* < 0.01; * *p* < 0.05.

**Figure 2 cells-13-00741-f002:**
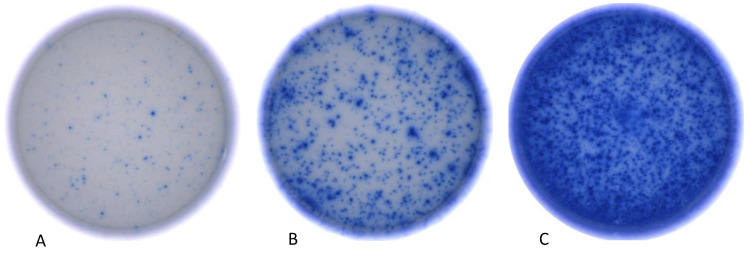
Photographic depiction of individual wells from the Human Interferon-γ Single Color Immunospot Assay, showcasing diverse levels of Interferon-γ activity as determined by the ImmunoSpot^®^ Reader from CTL (Cellular Technology Limited, Shaker Heights, OH, USA). The patients corresponding to the depicted wells received three (**A**), two (**B**), and one (**C**) immunosuppressive agent(s), respectively.

**Figure 3 cells-13-00741-f003:**
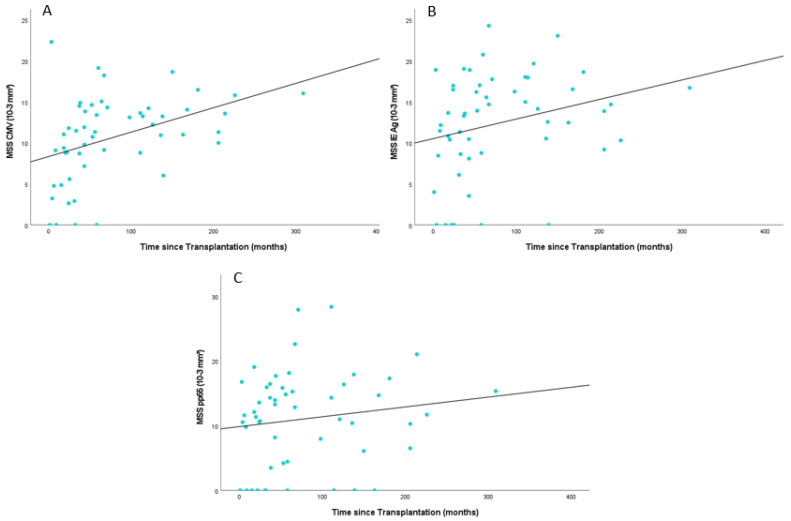
Scatter plot showing the mean spot size (MSS) of the individual antigens ((**A**) CMV peptide pool; (**B**) IE Antigen; (**C**) pp65) over time after transplantation; CMV = Cytomegalovirus, Ag = Antigen.

**Table 1 cells-13-00741-t001:** Comparison of patient characteristics in subgroups by number of immunosuppressive drugs. Analysis performed with descriptive statistics, Kruskal–Wallis, ANOVA, and Pearson Chi-Square tests; CMV = human cytomegalovirus; CMV D+/R serostatus = Donor status seropositive/Recipient status seronegative.

		One Immunosuppressant (*n* = 16)	Two Immunosuppressants (*n* = 27)	Three Immunosuppressants (*n* = 9)	*p*-Value
Patient characteristics	Age, years median (IQR)	70 (55.5–73.5)	54 (43–62)	51 (36.5–65)	0.051
	Gender, male (%)	6 (37.5)	14 (51.9)	7 (77.8)	0.18
	Months after Transplantation, median (IQR)	91 (57–157)	53 (24–136)	9 (3.5–40)	0.001
Pre-existing conditions	Diabetes mellitus (%)	5 (31.3)	10 (37)	1 (11.)	0.35
	Kidney insufficiency (%)	14 (87.5)	19 (70.4)	3 (33.3)	0.017
	Inflammatory disease (%)	5 (31.3)	5 (18.5)	1 (11.1)	0.44
Laboratory	Creatinine, mg/dL, median (IQR)	1.4 (1.21–1.83)	1.18 (0.85–1.43)	1.03 (0.91–1.24)	0.036
	Leucocytes, Tsd/µL, median (IQR)	5.1 (3.67–7.3)	5.5 (4.12–7.13)	6.05 (4.21–7.52)	0.89
	Positive CMV Serostatus before Transplantation	15 (93.8)	24 (88.8)	6 (66.6)	0.15
	CMV D+/R− serostatus at the time of blood collection	1 (6.3)	2 (7.4)	3 (33.3)	0.08
Medication	Tacrolimus (%)	7 (43.8)	19 (70.4)	7 (77.8)	0.15
	Everolimus (%)	6 (37.5)	5 (18.5)	4 (44.4)	0.27
	Mycophenolatmofetil (%)	2 (12.5)	24 (88.9)	8 (88.9)	<0.001
	Ciclosporin (%)	0 (0)	3 (11.1)	1 (11.1)	0.38
	Prednisolon (%)	0 (0)	1 (3.7)	7 (77.8)	<0.001
	Sirolimus (%)	1 (6.3)	1 (3.7)	0 (0)	1
	Valganciclovir	1	0	5	<0.001

**Table 2 cells-13-00741-t002:** Comparison of antigen detection within the subgroups of number of immunosuppressive drugs. Analysis performed using descriptive statistics.

	Detected Antigen Approaches (%)			
Number of Immunosuppressants	0	1	2	3
0	1 (14.3)	1 (14.3)	1 (14.3)	4 (57,1)
1	3 (18.8)	2 (12.5)	0	11 (68.8)
2	6 (22.2)	4 (14.8)	6 (22.2)	11 (40.7)
3	7 (77.8)	1 (11.1)	0	1 (11.1)

**Table 3 cells-13-00741-t003:** Odds Ratio and *p*-values for patient characteristics in the examined cohort, evaluated for each tested antigen using logistic regression analyses; CMV = human Cytomegalovirus.

	CMV Peptide Pool Detection		pp65 Detection		Early Antigen Detection	
	OR (95% CI)	*p*-Value	OR (95% CI)	*p*-Value	OR (95% CI)	*p*-Value
Age	0.944 (0.889–1.002)	0.06	0.932 (0.881–0.987)	0.016	0.958 (0.911–1.007)	0.091
Months after Transplantation	1.007 (0.994–1.019)	0.29	1.001 (0.991–1.011)	0.862	1.00 (0.991–1.010)	0.923
Renal failure	4.165 (0.777–22.326)	0.096	2.601 (0.497–13.604)	0.257	2.184 (0.479–9.96)	0.313
Tacrolimus	1.314 (0.302–5.713)	0.716	0.569 (0.138–2.344)	0.435	0.512 (0.132–1.993)	0.334
Prednisolone	0.065 (0.005–0.84)	0.036	0.048 (0.003–0.699)	0.026	0.144 (0.017–1.196)	0.073

**Table 4 cells-13-00741-t004:** Supplementary Analyses of CMV Reactivation and Immune Response Parameters; Statistical analyses employed the Mann–Whitney U Test and T-Test. CMV = human Cytomegalovirus, Ag = Antigen, D+/R− = Donor seropositive/Recipient seronegative.

	CMV Reactivation (*n* = 10)	No CMV Reactivation (*n* = 42)	*p*-Value
Age, median (IQR)	48.5 (39–65)	57.5 (48.8–70.0)	0.13
Gender, male (%)	5 (50)	22 (52.4)	0.89
Months after Transplantation median (IQR)	37.5 (8.25–95)	59 (29.5–128.5)	0.19
3 immunosuppressive agents, n (%)	6 (60)	3 (7.1)	<0.001
D+/R− CMV serostatus	4 (40)	2 (4.8)	0.002
Tacrolimus, n (%)	6 (60)	27 (64.3)	0.8
Mycophenolate mofetile, n (%)	8 (80)	26 (61.9)	0.28
Prednisolone, n (%)	6 (60)	3 (7.1)	<0.001
Sandimmun n (%)	1 (10)	3 (7.1)	0.76
Everolimus n (%)	3 (30)	11 (26.2)	0.39
Stimulation Index, median (IQR)			
CMV peptide Pool	0.5 (0.0–2.88)	37.3 (1.98–193.88)	<0.001
IE Ag	0.13 (0.0–1.85)	9.25 (1.9–42.57)	<0.001
pp65	0.61 (0.0–1.85)	13.5 (0.5–93.22)	0.004
Mean Spot Count, median (IQR)			
CMV peptide Pool	0.5 (0.0–27.38)	105.25 (4.5–267)	0.002
IE Ag	0.0 (0.0–0.0)	12.5 (0.0–113.13)	<0.001
pp65	1.75 (0.0–8.60)	20.88 (1.88–173.63)	0.012
Mean Spot size × 10^−3^ mm^2^, median (IQR)			
CMV peptide Pool	4.82 (0.0–10.49)	12.02 (9.03–14.37)	0.007
IE Ag	8.27 (0.0–11.55)	14.42 (10.5–17.8)	0.002
pp65	7.34 (0.0–12.02)	13.21 (7.50–16.53)	0.025

## Data Availability

The data presented in this study are available on request from the corresponding author due to patient´s privacy concerns.

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
