# Peer review of "Immunomonitoring via ELISPOT Assay Reveals Attenuated T-Cell Immunity to CMV in Immunocompromised Liver-Transplant Patients"

_cells, 2024, doi:10.3390/cells13090741_

Round 1
Reviewer 1 Report (Previous Reviewer 1)
Comments and Suggestions for Authors
By adding a healthy control cohort, the authors addressed my concerns about the study design.
Author Response
Please see the attachment.

Reviewer 2 Report (New Reviewer)
Comments and Suggestions for Authors
This study examined the anti-CMV T cell activity using ELISPOT assay for liver transplant patients. The authors showed that old age, immune-suppressive therapy using predonine, and early post-transplant period significantly reduced ant-CMV T cell immunity. Furthermore, patients with undetected CMV antigens in ELISPOT assay exhibited higher rate of CMV reactivation within early post-transplant phase. The authors emphasized the need for CMV reactivation monitoring using ELISPOT assay. This study indicates the significance of immune-monitoring for CMV, however, there are some drawbacks for publication.
1. The authors demonstrated that ELISPOT assay using CMV peptides, IE antigen, and pp65 is useful for monitoring CMV reactivation. In clinical, ELISPOT assay using all antigens is time-consuming and may be not applicable. Which antigen is the feasible for clinics?
2. Which is the most useful method for assessing of detection of antigen among Stimulation index, mean spot count, and mean spot size? Which methods is the most useful for detecting CMV reactivation?
3. The authors showed that prednisolone is associated with reduced antigen detection and CMV reactivation. However, prednisolone is mainly medicated in three immunosuppressive therapy, and not frequently used in two immunosuppressive therapy. The number of two immunosuppressive therapy consisting of calcineurin inhibitors and steroids is small. Steroids may not induce the reduced antigen detection.
4. The authors pointed that age is associated with anti-CMV T cell immunity. However, there were no data regarding the correlation of age and CMV reactivation. Did younger patients predispose to be associated with CMV reactivation? Was there association of post-transplant time with CMV reactivation?
Author Response
Please see the attachment.

Reviewer 3 Report (New Reviewer)
Comments and Suggestions for Authors
The authors intended to analyze T-cell immunity to CMV using the ELISPOT assay in patients with LT. Assessing immunity to CMV is essential in patients post-LT. The authors examined 52 patients by evaluating CMV-antigen-specific interferon-gamma secretion. The patients were assessed in different stages after LT, with varying immunosuppression regimens (specific to the distance after LT).
The authors must improve the abstract by presenting data (numbers, significance) regarding their most important results. They must specify the most important novelties brought by their study regarding the care of patients with LT. Some findings may be related to different recommendations of immunosuppression regimens following LT (regarding the number of drugs) or the presence of valganciclovir indications or prednisolone. These findings are also presented as limitations, so the study must probably be improved.
The images and graphs may be enlarged.
Comments on the Quality of English LanguageMinor corrections of the English language.
Round 2
Reviewer 3 Report (New Reviewer)
Comments and Suggestions for Authors
The authors imporved the abstract of the manuscript based on the recommendations. The study's limitations remain, and considering the analysis and discussion of those limitations and the possible overcome in the future, the manuscript may be considered for publication. The future studies should be made according to those limitations.
Comments on the Quality of English LanguageMinor editing is needed.
This manuscript is a resubmission of an earlier submission. The following is a list of the peer review reports and author responses from that submission.
Round 1
Reviewer 1 Report
Comments and Suggestions for Authors
With great interest, I have read the manuscript "Immunomonitoring via ELISPOT Assay Reveals Attenuated T-Cell Immunity to CMV in Immunocompromised Liver Transplant Patients“ by Traska and colleagues. In their study, the authors aim to characterize the HCMV-specific T cell responses at a single time point in a relatively small study cohort (N = 52) of liver transplant recipients. Among the main findings of their study, the authors state that the HCMV-specific T cell responses decrease with an increasing level of immune suppression. As a major concern, samples of patients treated with one immunosuppressant were collected significantly earlier compared to patients treated with two or three immunosuppressants. For me, it is thus very hard to determine whether the observed effects really depend on the varying immunosuppressive therapies. The authors need to address this major concern in their manuscript and include, for example, additional samples in their study. Additionally, I have additional minor comments:
Abstract:
Line 15-16 and 22-23: Both sentences seem grammatically incorrect.
Introduction:
Line 34: A HCMV-seroprevalence of 80% in Europe and North America seems to be very high.
Line 37-39: In rare instances, HCMV infections may cause severe infections in immunocompetent individuals.
Line 39: Typo in “t”
General comment: The authors may provide additional information about the role of the HCMV-specific CD8+ T cell response.
Material and Methods:
Information about the HCMV D/R serostatus is missing. Please add the number of HCMV R+ and D+/R- patients. In addition, information about the CMV prophylaxis and the schema of the CMV reactivation is missing (I assume the center uses a CMV qPCR). Please also add these data to the manuscript.
Table 1: The term “Leucocytes, Tsd/micromole” is probably wrong.
Results:
Fig.1 and corresponding text: MSS for IE and pp65 are missing. Is there a reason for that?
Table 2 and corresponding text: The given number of antigens detected is misleading, as the CMV peptide pool alone consists of five antigens.
Reviewer 2 Report
Comments and Suggestions for Authors
The manuscript by Traska et al investigates T-cell immunity against CMV, as measured by Elispot, in patients after liver transplantation. Several factors were identified as influencing T-cell immunity, including age and number of immunosuppressive drugs taken. A total of 52 patients were studied. The study and the results are interesting, although it has some shortcomings. For example, the patient cohort is relatively small and the varying responses to the different antigens weaken the significance of the results.
The following points should be noted:
- The authors note that the time since transplantation was a factor influencing the immune response. It is not clear whether this association is related to the fact that these patients may have reduced the number of immunosuppressive drugs later after transplantation and this may have caused this effect. An additional analysis to rule out this effect would be useful.
- The authors state that the results of the Elispot could (retrospectively) predict CMV reactivations. This is an interesting aspect, but needs to be further analyzed. Was CMV PCR routinely performed in all these patients? Or only in patients with an indication or with symptoms? Were the CMV reactivations clinically relevant, and if so, in what way?
- An interesting question that was not analyzed in this study would be the extent to which the results differ depending on the use of special immunosuppressants (e.g. tacrolimus vs. everolimus); if the data are available, an analysis of this would be useful.
- I recommend citation of the following papers:
https://pubmed.ncbi.nlm.nih.gov/33368987/
https://pubmed.ncbi.nlm.nih.gov/33504093/
https://pubmed.ncbi.nlm.nih.gov/30243982/
- The manuscript is reasonably well written, but it seems to need revision as some parts are not written with the necessary thoroughness. Some examples would be
a. Renal failure instead of renal insufficiency
b. A significance has been stated in Figure 1d, but this is not marked in the figure.
c. In section 3.5, proper names of drugs (Prograf, Decortin) are given, which should be changed.
d. The first paragraph of the discussion is much too long and therefore not easy to read.
Comments on the Quality of English Language
I have already made some comments on this topic in the paragraph above.
Reviewer 3 Report
Comments and Suggestions for Authors
Immunosuppressive drugs are necessary to prevent acute rejection in organ transplant recipients but inevitably increase the risk of opportunistic infections. In this manuscript, Cells 2822683, the authors analyzed the relationship between immunosuppressive therapy and ELISPOT score against CMV in liver transplanted recipients. These patients received single, dual, or triple immunosuppressive drug therapy. ELISPOT scores decreased with increasing immunosuppressive drugs. Furthermore, the decrease of ELISPOT response to CMV antigen peptides was highly correlated to the CMV reactivation. Since CMV is one of the major pathogen of opportunistic infection, these findings are meaningful to improve the anti-CMV treatment in transplanted recipients. However, there is some bias in the post-transplantation period, as increased immunosuppressive drugs shorten the time after transplantation. Authors should clarify following points.
Specific comments
lines 55-69
PBMCs were collected from 52 CMV seropositive transplanted patients in this report. It should be described that the serostatus before transplantation. Additionally, was the CMV genome detectable in collected PBMCs?
Line 65
Cryopreserved PBMCs were used in this report. Generally, the cryopreserve process reduces the cell reactivity to antigens. Authors should discuss the impact of cryopreserve processes on the results of ELISPOT assay.
Lines 86
Regarding the origin of peptides; pp65 and IE peptides are derived from the AD169 strain and Towne strain, respectively. Are these peptides unique to these strains? As the strain specific epitopes are known in CMV, it is possible that individual patients are infected with different CMV strains and present different immune responses. Thus, it is necessary to discuss that the impact of the difference of peptide sequence to immune response. Furthermore, the origin of other peptides should be shown.
Lines 151-152
Patents received single, dual, or triple immunosuppressive therapies. Furthermore, about 20% of patients were received steroid treatment. Authors should describe the reason why these patients treated different immunosuppressive drugs.
Table 1
Please add the history of anti-CMV drugs and immunoglobulin treatments. It is necessary to clarify whether the treatment of antiviral drugs prevented the reactivation in patents receiving three immunosuppressants.
Table 1 and lines 213-222
The significant difference in “Month after Transplantation, median” were observed among these subgroups; 91, 53, and 9 months after transplantation in one, two, and three immunosuppressants subgroups, respectively. In lines 213-222, authors described the correlations between the mean spot size of ELISPOT and time after transplantation. This could be confounding, i.e. the patents, who received three immunosuppressants and had a shorter time since transplantation, were highly reduced cell associated immunity. Authors should describe the analyses within subgroups.
Lines 213-234
Authors showed that the mean spot size of ELISPOT assay was increased by the time since transplantation. Please indicate the number of specimens in each group. Furthermore, to help understandings, it is better to plot the relationship between the time since transplantation and the spot size as Figure 3.
Minor comments
Lines 39, 59, and others
In this manuscript, cytomegalovirus is abbreviated as hCMV, CMV, or HCMV. Authors should standardize these abbreviations.
Line 39
typos: t
Line 46
typos: Both “INF-ɤ” and “IFN-ɤ” are presented in this manuscript.
Line 130 and others
Although authors described “cytokine release”, “cytokine secretion”, and “cytokine production” in this manuscript, only IFN-ɤ was measured. Therefore, it is better to describe as “IFN-ɤ” unless there is a specific intention.
Figure 1
What does horizontal axis indicate?
Tables
typos: prednisolone
Round 2
Reviewer 1 Report
Comments and Suggestions for Authors
The authors tried to address my major concern. Although Fig. 3 is neither described nor referenced in the main text, I interpret that the time since transplantation has a substantial impact on the HCMV-specific cellular immune responses. The study cohort is thus, in my opinion, not suitable to address the main questions of the study. Therefore, I recommend rejecting the study.
Reviewer 2 Report
Comments and Suggestions for Authors
The authors have implemented the suggested points satisfactorily, so that the manuscript is clearly improved and, in my opinion, can be considered for publication.
Reviewer 3 Report
Comments and Suggestions for Authors
The revised version of cells-2822683 was well revised in accordance with the reviewer’s comments. Thus, this manuscript is acceptable by “Cells”.
But generally we abbreviate human cytomegalovirus as HCMV or CMV but not hCMV. I think "CMV" is more appropriate unless you have some special meanings.